# Towards multi-sequence MR image recovery from undersampled k-space data

**Cheng Peng**                                                                       cp4653@umd.edu
**Wei-An Lin**                                                                        walin@umd.edu
**Rama Chellappa**                                                          rama@umiacs.umd.edu
*University of Maryland, College Park, USA*

**S. Kevin Zhou**                                                    s.kevin.zhou@gmail.com
*Chinese Academy of Sciences*
*Peng Cheng Laboratory, Shenzhen, China*

**Editors:** Under Review for MIDL 2020

## Abstract

Undersampled MR image recovery has been widely studied with Deep Learning methods as a post-processing step for accelerating MR acquisition. In this paper, we aim to optimize multi-sequence MR image recovery from undersampled k-space data under an overall time constraint. We first formulate it as a *constrained optimization* problem and show that finding the optimal sampling strategy for all sequences and the optimal recovery model for such sampling strategy is *combinatorial* and hence computationally prohibitive. To solve this problem, we propose a *blind recovery model* that simultaneously recovers multiple sequences, and an efficient approach to find proper combination of sampling strategy and recovery model. Our experiments demonstrate that the proposed method outperforms sequence-wise recovery, and sheds light on how to decide the undersampling strategy for sequences within an overall time budget.

**Keywords:** Magnetic Resonance, Image Recovery, Multi-Modal

## 1. Introduction

Magnetic Resonance Imaging (MRI) is a widely used medical imaging technique. It holds several distinct advantages over other imaging modalities such as computed tomography (CT) and ultrasound. Not ony does MRI resolve tissues at a high quality, it can also be customized with different pulse sequences to produce a variety of desired contrasts that reveal different kinds of tissues, such as blood vessels and tumor regions. Furthermore, compared to CT, MRI does not expose patients to ionizing radiation. On the other hand, MRI is limited by its long acquisition time, as the data is acquired by traversing through k-space, where the speed of traversal is limited by the underlying MR physics and machine quality. In practice, patients often take multiple MR sequences, each of which uses different parameters to target specific tissues and lesions, resulting in even longer overall acquisition time. This leads to various practical problems, ranging from image blurriness due to patient movement to limiting accessibility of the machines.

There is a long history of research on how to undersample MR k-space data while maintaining image quality. Lustig et al. (Lustig et al., 2007) first proposed to use Compressed

Sensing in MRI (CSMRI), assuming that the undersampled MR images have a sparse representation in some transform domain, where noise can be discarded through minimizing the $\mathcal{L}_0$ norm of such representation. This method was shown to yield much better results than zero-filling the missing k-space samples (ZF); Extending on CSMRI, Ravishankar et al. (Ravishankar and Bresler, 2011) applied more adaptive sparse modelling through Dictionary Learning, where the transformation is optimized through specific sets of data, resulting in better sparsity encoding. To further explore redundancy within the MR data, new methods have been proposed in recent years (Huang et al., 2012; Hirabayashi et al., 2015; Senel et al., 2019; Gozcu et al., 2018; Gong et al., 2015), focusing on extrapolating information in adjacent slices, in multi-acquisition scenarios, and in scenarios where additional sequence is available. In the domain of Deep Learning, Schlemper et al. (Schlemper et al., 2018) proposed a cascade of CNNs that incorporates data consistency layers to de-noise MRI in image domain while maintaining consistency in the k-space, and showed that the results significantly outperformed DLMRI (Ravishankar and Bresler, 2011). Yang et al. (Yang et al., 2018) proposed DAGAN, which recovers undersampled MR images through a U-Net structure with perceptual and adversarial loss in addition to $L_1$ loss in image space and frequency space. Quan et al. (Quan et al., 2018) proposed RefineGAN, which performs reconstruction and refinement through two different networks, and enforces a cyclic loss in the image and frequency spaces.

Although the mentioned CNN-based methods have obtained impressive results, they focus on single sequence reconstruction. Few studies have explored the effectiveness of CNN-based methods under multi-sequence scenarios, which are common in practice and shown to contribute in non-learning-based methods (Gong et al., 2015; Bilgic et al., 2018). Xiang et al. (Xiang et al., 2018) showed that a highly undersampled $T_2$ sequence, given a fully sampled $T_1$ sequence, can be well recovered through a Dense U-Net. In this paper, we attempt to find the best strategy at undersampling k-space acquisition over multiple sequences, such that we can best recover the sequences post-acquisition. Wang et al.(Wang et al., 2017) also explored the feasibility of multi-constrast MR imaging through CNN models.

The contributions of our paper can be summarized as follows: (i) we formulate a *combinatorial constrained optimization* problem, where given a limited acquisition time, we seek to find the best strategy to undersample the k-spaces of multiple sequences to achieve the best overall recovery; (ii) we propose a novel CNN-based *blind recovery model* that extrapolates the shared information across different sequences and simultaneously recover them, as well as an efficient approach to finding a proper combination of sampling strategy and recovery model; (iii) we perform extensive evaluation on real and simulated k-space data, which shows that the proposed model outperforms the method of independently recovering each sequence, and that our method finds *the undersampling strategy adaptive to the given sequences*.

## 2. Problem Formulation

We first note that the most popular MR k-space sampling method is through Cartesian trajectory, where a series of acquisitions is performed along equally-spaced parallel lines, which are conventionally called *phase encoding lines*. This leads to a natural implementation

for MR undersampling, where the technicians can drop certain phase encoding lines from the sampling grid (Lustig et al., 2007). In this paper, we focus on undersampling with 1D masks along the phase encoding direction[1].

Consider multiple MR sequences with full k-space spectrums $\{F_s\}_{s=1}^S$, where $S$ denotes the total number of sequences, with each spectrum sampled by $N$ phase encoding lines. For each $F_s$, the unit time for sampling a phase encoding line is denoted by $t_s$. We define 1D sampling masks $\mathcal{M}_s \in \{0,1\}^N$ which selects a subset of encoding lines $\mathcal{M}_s \odot F_s$ for faster acquisition. By applying the inverse Fourier transform $\mathcal{F}^{-1}$, an undersampled MR image for sequence $s$ is reconstructed as

$$I_{M_s} = \mathcal{F}^{-1}(\mathcal{M}_s \odot F_s). \tag{1}$$

When fully sampled, the MR image is reconstructed by $I_s = \mathcal{F}^{-1}(F_s)$. If we denote the number of selected encoding lines by $|\mathcal{M}_s|$, the total time needed to acquire all the sequences is $T = \sum_{s=1}^S t_s \times |\mathcal{M}_s|$.

Undersampled MR leads to faster acquisition and degraded quality compared to fully sampled MR. To allow fast acquisition while retaining image quality, we apply a deep neural network as the post-processing step to improve the degraded image quality. Therefore, we consider the problem of searching for an optimal sampling strategy $\{\mathcal{M}_s\}_{s=1}^S$ and a CNN $f_\theta$ that best recovers fully sampled $\{I_s\}_{s=1}^S$ from $\{I_{\mathcal{M}_s}\}$ with a time constraint $T \leq T_{max}$. This constrained optimization problem can be formulated as follows:

$$\min_{\theta,\{\mathcal{M}_s\}} \sum_{s=1}^S E_{I_s \sim p(I_s)} \left[ \left\| f_\theta(I_{\mathcal{M}_s}) - I_s \right\|_1 \right] \quad \text{s.t.} \quad \sum_{s=1}^S t_s |\mathcal{M}_s| \leq T_{max}. \tag{2}$$

We use the $L_1$ loss in (2); however, other loss functions can be used too.

The problem defined in (2) is *combinatorial* in nature, as has been realized by Reeves et al. (Reeves and Heck, 1995). First, the set $\{\mathcal{M}_s\}_{s=1}^S$ has a total of $2^{NS}$ possible combinations. Secondly, the best recovery model depends on the choice of sampling strategy. As a result, the optimal solution to (2) is in general difficult to find. As a preliminary attempt, we assume a fixed candidate set $\mathcal{C} \in \{m_1, \ldots, m_C\}$ for each $\mathcal{M}_s$. The number of possible sampling strategies becomes $C^S$ instead. However, even with the simplification, a straightforward approach to (2), which is

$$\min_{\mathcal{M}_{1:S} \in \mathcal{C}^S} \left( \min_\theta \sum_{s=1}^S E_{I_s \sim p(I_s)} \left[ \left\| f_\theta(I_{\mathcal{M}_s}) - I_s \right\|_1 \right] \right) \quad \text{s.t.} \quad \sum_{s=1}^S t_s |\mathcal{M}_s| \leq T_{max}, \tag{3}$$

still requires training $C^S$ models and then choosing the one with minimum loss. This is necessary since each model is trained to best eliminate noise introduced by the specific $\mathcal{M}_s$, and becomes sub-optimal when the noise level/pattern is changed.

In this work, we propose an efficient approach that finds a $(\theta, \{\mathcal{M}_s\}_{s=1}^S)$ while circumventing the computational cost in training an excessive number of models. Conceptually, we propose to first train a blind recovery model (BRM), which takes randomly undersampled

---

1. We have found that undersampling with 2D masks generally leads to better recovery quality; however, such a setting is less time efficient in practice.

MR sequences as inputs, and recovers them to fully sampled MR sequences. The trained BRM can then be used as an MR sequence quality estimator to search for the optimal sampling strategy $\{\mathcal{M}_s^*\}_{s=1}^S$. Finally, with $\{\mathcal{M}_s^*\}_{s=1}^S$, we can proceed to solve (3) by fine-tuning on the existing BRM. In total, the proposed method only requires training *one* CNN, which significantly reduces the computational cost.

### 2.1. Blind recovery model

A blind recovery model (BRM) is a CNN $f_\theta$ which recovers $I_s$ by fusing information from different undersampled MR sequences $\{I_{\mathcal{M}_s}\}_{s=1}^S$, $\mathcal{M}_s \in \mathcal{C}$. We adopt a data augmentation approach, which randomly selects sampling masks from $\mathcal{C}$, and consider the following *unconstrained optimization problem*:

$$\theta^* = \arg\min_\theta \sum_{s=1}^S E_{I_s \sim p(I_s), \mathcal{M}_s \sim p(\mathcal{C})} \left[ \left\| f_\theta(I_{\mathcal{M}_s}) - I_s \right\|_1 \right]. \tag{4}$$

As we will show, the model trained under this scheme sacrifices its ability to fit on a specific sampling profile, and in exchange performs generally well across all sampling profiles. Therefore, it can serve as a good estimator for discovering the best sampling strategy.

### 2.2. Sampling strategy searching

Given a trained BRM $f_{\theta^*}$, we propose to search for the optimal sampling strategy by finding the one with a minimum loss:

$$\mathcal{M}_{1:S}^* = \arg\min_{\mathcal{M}_{1:S}} \sum_{s=1}^S E_{I_s \sim p(I_s)} \left[ \left\| f_{\theta^*}(I_{\mathcal{M}_s}) - I_s \right\|_1 \right] \text{ s.t. } \sum_{s=1}^S t_s |\mathcal{M}_s| \le T_{max}. \tag{5}$$

The above exhaustive search requires $C^S$ forward passes, which is significantly less computationally heavy than training $C^S$ CNNs. The solution $\theta^*$ can be further improved by learning a refined model specific to $\mathcal{M}_s^*$:

$$\hat{\theta} = \arg\min_\theta \sum_{s=1}^S E_{I_s \sim p(I_s)} \left[ \left\| f_\theta(I_{\mathcal{M}_s^*}) - I_s \right\|_1 \right]. \tag{6}$$

### 2.3. Single sequence training vs multi-sequence training

One has the option of training (a) multiple SISO (single input single output) BRMs, one per sequence, or (b) one monolithic MIMO (multiple input multiple output) BRM for all sequences. The latter option holds several advantages over the former. First, option (a) does not consider the complementary information across different sequences. As shown in (Xiang et al., 2018; Huang et al., 2012), there exists a strong correlation between sequences of the same patient, as they share the underlying anatomical structures. If a particular sequence is severely undersampled, leading to the loss of some anatomical detail, such information may be present in other less severely undersampled sequences. Secondly, option (b) only requires training one model, while option (a) requires $S$ models. As all the models attempt to eliminate distortions due to undersampling, they should learn similar features. However, multi-sequence training requires the sequences to be aligned amongst themselves, which may require coordination with the patient or proper registration algorithms.

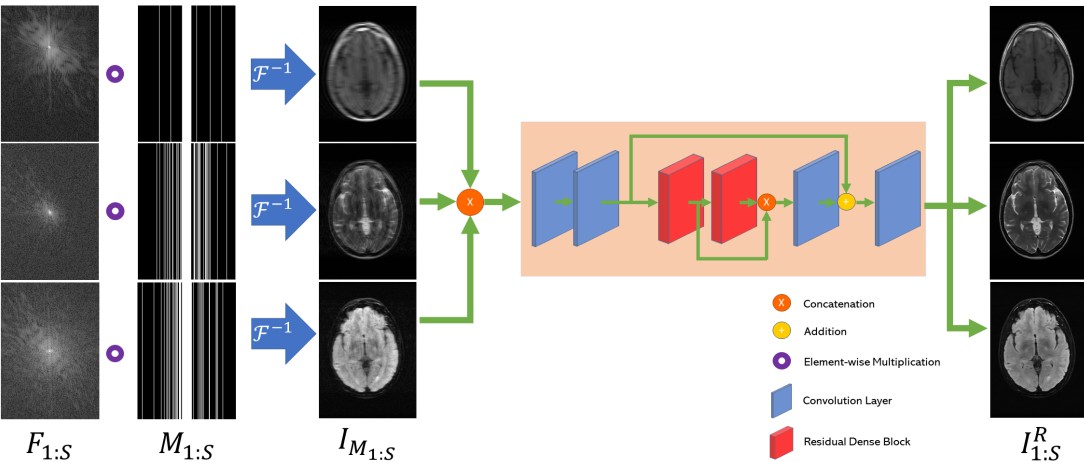

$F_{1:S}$ $\quad\quad$ $M_{1:S}$ $\quad\quad$ $I_{M_{1:S}}$ $\quad\quad\quad\quad\quad\quad\quad\quad\quad\quad\quad\quad\quad\quad\quad$ $I^R_{1:S}$

Figure 1: Multi-sequence recovery (MIMO) pipeline with the masks $M_s$ randomly selected. SISO pipeline is implemented similarly with a single sequence and a single output.

## 2.4. Network architecture

Our multi-sequence simultaneous recovery approach is shown in Fig 1. The approach is based on Residual Dense Block (RDB) (Zhang et al., 2018), which incorporates the idea of residual learning and dense block (Huang et al., 2016), allowing all layers of features to be seen directly by other layers. During learning, each raw k-space data $F_s$ first gets undersampled through a randomly generated mask $\mathcal{M}_s$. The results are then transformed from k-space to image space, and concatenated before sent to the recovery network, which outputs $I^R_{1:S}$. The loss function is defined as the following: $\mathcal{L} = \|I^R_{1:S} - I_{1:S}\|_1$.

## 3. Experiments

### 3.1. Datasets

We employ two datasets. The first one is a privately collected, k-space raw data of three sequences ($T_1$, $T_2$, FLAIR) from 20 patients, with each sequence containing 18 slices. The sequences are co-registered and taken with an MRI machine with 8 channels; in order to augment training, we treat each channel as an individual image to result in a total of 2,880 three-sequence images, which are divided into a ratio of 17:1:2 for training, validation, and testing. We refer to this dataset as "real data". In order to further validate our research, we also employ the Brain Tumor Image Segmentation (BraTS) dataset (Menze et al., 2015; Bakas et al., 2017), which contains $T_1$, $T_2$, and FLAIR. The sequence are co-registered to the same anatomical template, skull-stripped, and interpolated to the same resolution. We divide the selected 167 cases into a ratio of 140:10:17 for training, validation, and testing. From every case, we select the middle 60 slices that contain most of the anatomical details. Because BraTS does not provide raw k-space data, we follow common practices (Xiang et al., 2018; Yang et al., 2018) to simulate k-space data. We refer to this dataset as "simulated data". We implement the proposed approach using PyTorch and train all the models with

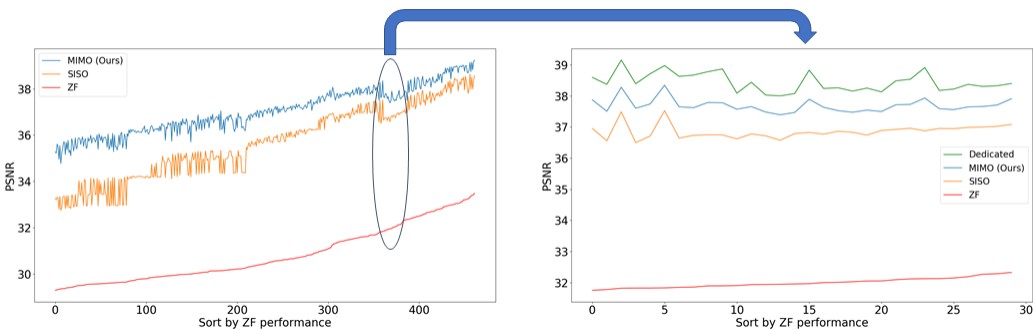

Figure 2: Quantitative recovery performance comparison. The Pearson correlation coefficient between Dedicated and MIMO vs between Dedicated and ZF is 0.85 vs -0.33 in the selected range

Adam. *Below, our insights are first demonstrated with experiments on real data and are further validated on simulated data.* optimization, a momentum of 0.5 and a learning rate of 0.0001, until they reach convergence.

### 3.2. Acquisition time and undersampling settings

In general, $T_2$ and FLAIR have a longer repetition time (TR) than $T_1$; however, the acquisition time of each sequence also depends on the number of excitations. A larger number of excitations helps better resolve sequences but take a longer time. Therefore, the acquisition time of each sequence is rather machine-dependent. Here we consider three experimental settings: $t_{T_1}$:$t_{T_2}$:$t_{flair}$= (1) 1:1:1, (2) 1:4:6, and (3) 2:3:6.

We experiment on both low-pass sampling (Xiang et al., 2018) and random sampling (Yang et al., 2018). We found that random sampling works better on real data but worse on simulated data. As our approach is agnostic of sampling strategy, we choose the better performing sampling strategy for each dataset. During BRM training, the masks $\mathcal{M}_{1:S}$ are generated based on a random $\lambda_s \in [1, k]$, where $k$ is the maximum undersampling factor (we set $k = 8$). This means that BRM, after training, can handle a continuous set of undersampling factors on every sequence.

### 3.3. Evaluation metrics

We utilize two metrics to gauge image quality: PSNR (peak signal-to-noise ratio) and SSIM (structural similarity). Since we mainly focus on three sequences, calculation of these metrics on three-sequence outputs is the same as on RGB images. This is easily extensible with a larger number of sequences. Since MRI images do not have a fixed dynamic range, PSNR values should be regarded for their relative improvements.

### 3.4. Main results

We evaluate the effectiveness of BRM to empirically prove that a properly trained network $f_\theta$ performs well regardless of the choices of $\mathcal{M}_{1:S}$, and serves as a good estimator of best

sampling strategy. Furthermore, we show that MIMO BRM performs better than SISO BRM.

| $t_{T_1}{:}t_{T_2}{:}t_{flair}$ | $\lambda_{T_1}, \lambda_{T_2}, \lambda_{flair}$ | ZF | SISO | MIMO | MIMO (tuned) |
|---|---|---|---|---|---|
| 1 : 1 : 1 Real | 6.6, 2.1, 8.0 | 33.48/0.918 | 38.57/0.980 | 39.24/0.984 | 40.00/0.987 |
| | 8.00, 2.11, 6.63 | 33.43/0.920 | 38.36/0.979 | 39.16/0.984 | **40.07/0.987** |
| | 7.25, 2.11, 7.25 | 33.39/0.918 | 38.50/0.980 | 39.15/0.984 | 40.07/0.986 |
| 1 : 4 : 6 Real | 2.90, 2.44, 7.82 | 33.81/0.926 | 38.85/0.983 | 39.33/0.985 | 40.28/0.988 |
| | 3.01, 2.44, 7.69 | 33.60/0.924 | 38.83/0.983 | 39.32/0.985 | **40.37/0.987** |
| | 3.93, 2.44, 6.99 | 33.58/0.925 | 38.81/0.983 | 39.31/0.986 | 40.13/0.987 |
| 1 : 1 : 1 Simulated | 5.66, 3.14, 3.93 | 32.21/0.887 | 37.69/0.974 | 38.32/0.978 | **38.99/0.980** |
| | 5.27, 3.41, 3.74 | 32.31/0.889 | 37.88/0.975 | 38.31/0.979 | 38.98/0.980 |
| | 6.10, 3.14, 3.74 | 32.21/0.887 | 37.51/0.973 | 38.31/0.978 | 38.99/0.980 |
| 2 : 3 : 6 Simulated | 2.61, 3.74, 5.16 | 32.87/0.899 | 38.01/0.976 | 38.67/0.980 | **39.37/0.982** |
| | 2.44, 3.74, 5.40 | 32.84/0.899 | 37.87/0.975 | 38.66/0.980 | 39.35/0.982 |
| | 2.61, 3.41, 5.66 | 32.82/0.899 | 37.80/0.975 | 38.65/0.980 | 39.33/0.982 |

Figure 3: Quantitative evaluations for the top performing $\lambda_{1:S}$ under different acquisition time assumption. The performance numbers presented here are PSNR (dB) and SSIM.

The study is done by training (i) a MIMO BRM, (ii) three SISO BRM, one for every sequences, and (iii) many models that are dedicated for specific sampling ratios. All the models follow the same structure shown in Fig. 1. The proposed training scheme for continuous $\lambda_s \in [1, k]$ allows us to efficiently investigate the performance of different undersampling strategies. For each acquisition time setting $\{t_s\}_{s=1}^{S}$, we search through possible $\{\lambda_s\}_{s=1}^{S}$ on the following simplex: $\sum_{s=1}^{S} \frac{t_s}{\lambda_s} = T_{max}$, which maximally utilizes the budgeted time $T_{max}$. We select hundreds of $\{\lambda_s\}_{s=1}^{S}$ under the 1:1:1 time setting, and set $T_{max} = \frac{T}{4}$, or 75% reduction in time. We run the trained models on the test set, and plot the reconstruction performances in Fig. 2. The top-three performing sampling strategies for different acquisition time setting are shown in Table 3.

Fig. 2 shows a clear performance gap between MIMO and SISO. Overall, the reconstruction performance of ZF images is positively correlated with the performances of BRMs; however, the correlation fluctuates often, and two sets of ZF that are similar in PSNR can swing for more than 1dB after the images are processed through BRM. To limit the number of dedicated models we need to train, we select a range of sampling factors of which ZF performance does not correlate well with MIMO/SISO performance, and train 30 dedicated models to see how well BRM predicts the performance of dedicated models. As we observe from the right image in Fig. 2, our BRM, both from MIMO and SISO settings, predicts the performance of dedicated models with a high correlation. We further choose the best three $\{\lambda_s\}_{s=1}^{S}$, and perform the last stage of fine-tuning accordingly to (6). A visual evaluation on real data is shown in Fig. 4. For more visual results, please refer to the Supplemental Material section.

Base on the best performing $\{\lambda_s\}_{s=1}^{S}$, we perceive that among $T_1$, $T_2$, and FLAIR, the results are best when $T_2$ is sampled the most. We suggest that this makes intuitive sense

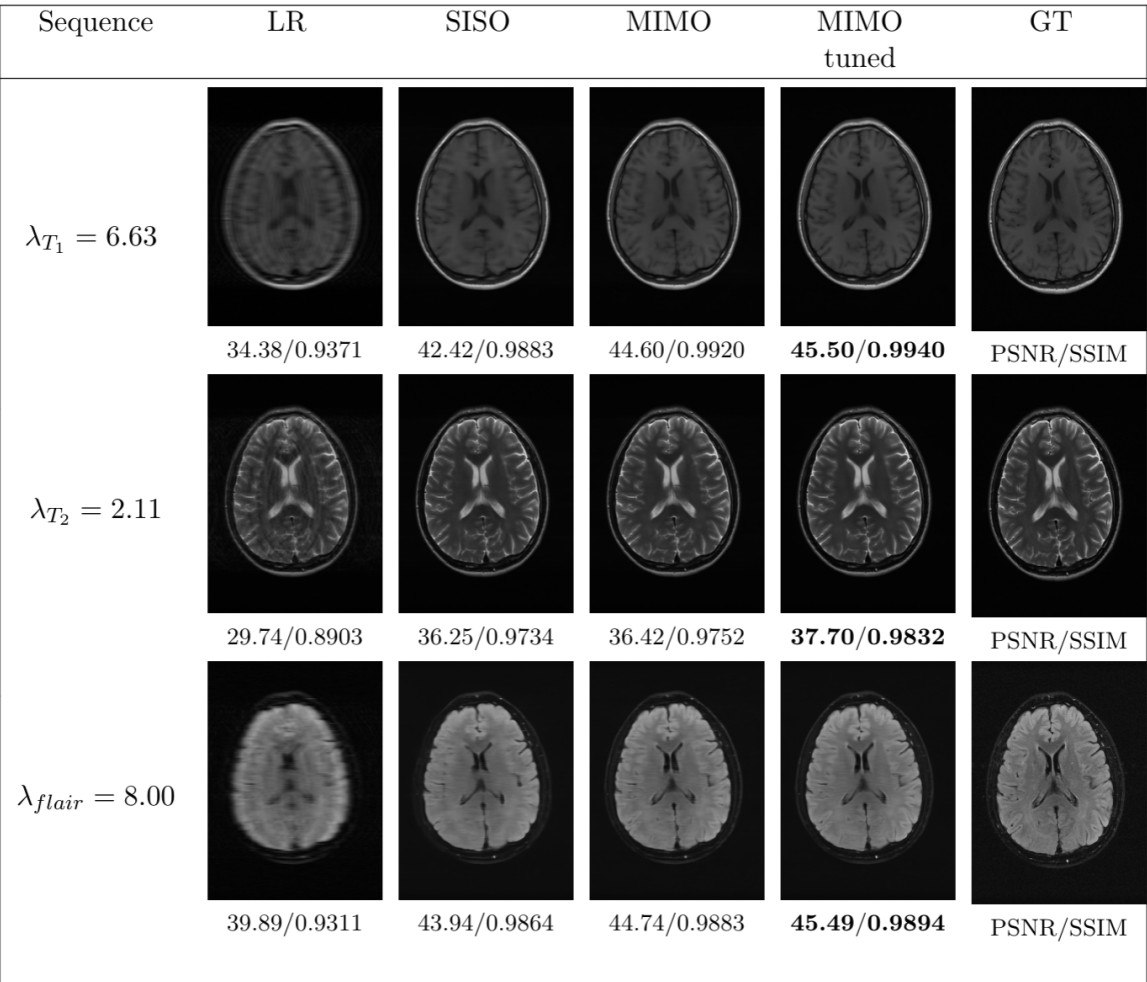

| Sequence | LR | SISO | MIMO | MIMO tuned | GT |
|---|---|---|---|---|---|
| $\lambda_{T_1} = 6.63$ | 34.38/0.9371 | 42.42/0.9883 | 44.60/0.9920 | **45.50/0.9940** | PSNR/SSIM |
| $\lambda_{T_2} = 2.11$ | 29.74/0.8903 | 36.25/0.9734 | 36.42/0.9752 | **37.70/0.9832** | PSNR/SSIM |
| $\lambda_{flair} = 8.00$ | 39.89/0.9311 | 43.94/0.9864 | 44.74/0.9883 | **45.49/0.9894** | PSNR/SSIM |

Figure 4: Visual comparison of different methods, with PSNR (dB) and SSIM values listed under the images. After recovery, the images are sharper with more visible details. Please refer to Fig. **??** in Supplemental Material for the respective difference maps against HR.

as $T_2$ images provide the best contrast out of the three sequences, which can compensate for the details lost in other images. The same observation can be made on the simulated data, where both $T_2$ and FLAIR show good contrast. When the time setting is changed to non-uniformity, we can see that our search for the best sampling strategy reflects the change. $T_1$ is sampled more as a result of faster acquisition time, while $T_2$ is still sufficiently sampled.

## 4. Conclusion

In this work, we formulated multi-sequence MR recovery as a constrained optimization problem, and explored possible methods to solve such a problem. We proposed a CNN-

based approach and an optimization scheme that helps us find the proper combinations of sampling strategy and recovery model without combinatorial complexity. We evaluated our approach on both private raw data and public simulated data, demonstrating that our method can quickly finds the sampling strategy that yields superior reconstruction performance. We showed that our model outperforms single sequence recovery methods in terms of recovery quality, time and space complexity. We believe that our method, in combination with guidance from radiologists, can help reduce the acquisition time for multi-sequence scenarios.

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
