# OpenReview forum: "Towards multi-sequence MR image recovery from undersampled k-space data"
_MIDL.io/2020/Conference — MIDL 2020_

### Official Review · AnonReviewer3 · 2020-03-05
**Blind Recovery Model for multi-sequence MR reconstruction and optimal sampling pattern selection**

**Rating:** 2
**Confidence:** 4

**Summary:**

The paper formulates the problem of  multi-sequence MR reconstruction as a MR acquisition-time constrained optimization problem. The authors propose a blind recovery model (BRM) to discover the optimal sampling masks across sequences. Essentially the method consists in training the model using a set of random sampling masks. Then, running prediction across what I assume to be either the train or validation set to select the  sampling mask that gives the best results. Finally, the model is fine-tuned using the sampling mask selected in the previous step.

The choice of the optimal sampling mask using deep learning is a very important topic, but relatively unexplored. Therefore, I congratulate the authors on their efforts.

**Strengths:**

Long MR acquisition times limit the access of this exam to subjects in need of the exam. Therefore, it is really interesting that the authors included a time constraint to the MR reconstruction mathematical formulation. The authors propose a blind recovery model (BRM), which is well explained in the paper, to tackle this optimization problem. The BRM model seeks to optimize multi-sequence reconstruction while at the same time selecting the proper sampling masks.

**Weaknesses:**

The BRM does not guarantee the optimality of the results. In fact, it was expected that fine-tuning a model to a specific sampling mask would improve reconstruction. It was also expected that a MIMO model would outperform a SISO model, since this has previously been done.



**Justification Of Rating:**

Although I think it is a great idea to formulate the problem of  multi-sequence MR reconstruction as a MR acquisition-time constrained optimization, I believe the experiments shown in this paper are not sufficient to convince that BRM models will select the optimal sampling strategy  that falls within the time constraint.

**Paper Type:**

methodological development

**Questions To Address In The Rebuttal:**

- If you created one fine-tuned model for each of the $C^S$ sampling strategies, would the optimal sampling strategy still be the one the BRM model selected during the  $C^S$ forward passes? Can the authors include that experiment? Perhaps not with all  $C^S$ sampling strategies, but selected few percentiles (0%, 25%, 50%, 75%, 90%, 95%).

- Is the idea to train one model for each different multi-sequence acquisition protocol (T1/PD, T1/T1-e, T1/FLAIR,...) and for each time constraint established? Wouldn't that lead to a prohibitive number of models?

- Not sure why the authors are using channel-wise reconstruction. Are the metrics reported computed channel-wise as well? or the channels are somehow combined to produce a final reconstruction?

**Special Issue:**

no

---

> ### Author Response · Authors · 2020-03-26
> **Discussion/Rebuttal for AnonReviewer3**
>
> We thank the reviewer for the comments, and would like to address some of the concerns and provide some clarifications.
>
> - If you created one fine-tuned model for each of the  sampling strategies, would the optimal sampling strategy still be the one the BRM model selected during the   forward passes? Can the authors include that experiment? Perhaps not with all   sampling strategies, but selected few percentiles (0%, 25%, 50%, 75%, 90%, 95%).
>
> There seems to be some misunderstanding here. For fair comparisons, we compared the performance of BRM to fine-tuned models trained under specific sampling strategies. We showed in Figure 2 (on the right) that while there is a performance gap between the two methods, BRM is very indicative of the recovery quality of fine-tuned models (named 'Dedicated' in the graph). Instead of selecting from different percentile bins, we concentrated on a region where the zero-filled reconstruction quality is not well correlated with the CNN reconstruction quality (shown in the circled region in Figure 2, on the left). We trained ~30 models for different sampling strategies to demonstrate the empirical effectiveness of BRM in selecting sampling strategy.
>
> Due to the virus pandemic, we are unable to run the extra experiments reviewer proposed. However, since the ~30 models that we did train should very high correlation over an arbitrary range, there does not seem to be sufficient reasons to say the fine-tuned models would not correlate well otherwise.
>
> - Is the idea to train one model for each different multi-sequence acquisition protocol (T1/PD, T1/T1-e, T1/FLAIR,...) and for each time constraint established? Wouldn't that lead to a prohibitive number of models?
>
> In this paper, we look at the instance where the multi-sequence acquisition protocol is pre-determined (e.g. T1/T2/FLAIR) and try to determine how much sampling time should be distributed for different sequences, such that the overall recovery is the best (measured visually and through standard quantitative measurements). The reviewer pointed out another complexity at constructing the multi-sequence acquisition protocol, which is unfortunately not within the scope of this paper. Extensions of our work may be able to address this issue, e.g. by constructing a CNN to take all relevant sequences, and at training time mask out the sequences that are not acquired, which is a generalized version of our current method. Further experiments are needed to test the effectiveness of this extension.
>
> - Not sure why the authors are using channel-wise reconstruction. Are the metrics reported computed channel-wise as well? or the channels are somehow combined to produce a final reconstruction?
>
> We thank the reviewer for pointing the use of channel-wise reconstruction. Due to page limitation we refrained from elaborating on this choice. Due to the relatively small k-space data size, we use each channel as individual measurements/images to augment our dataset. This is valid both conceptually, as CNNs are translation-invariant, and empirically, as the final reconstruction shows results that are not biased towards any channel. The metrics and images are measured and compared based on the final reconstruction.

---

### Official Review · AnonReviewer2 · 2020-03-11
**This paper introduces a novel method for jointly estimation of multimodal MRI using under-sampled k-space data.**

**Rating:** 4
**Confidence:** 5
**Recommendation:** Oral

**Summary:**

This paper introduces an imaging acquisition and reconstruction framework for T1, T2 and FLAIR images. The k-space of the three modalities have complementary information. The undersampled image data were jointly used as input to a convolutional neural networks to simultaneously reconstruct high-quality images for the three modality. This method could significantly reduce the scan time in practice.

**Strengths:**

The introduced method unsamples the k-space of three modalities using different strategies. It then uses a MIMO CNN algorithm to simultaneously enhance image quality. This method could significantly reduce the scan time.
The performance was evaluated and compared with several other method and the ground truth.


**Weaknesses:**

The paper is quite self-contained with very convincing method and novelty. To further improve this paper, it is better to show the performance with tumor data, although it was mentioned that the BraTS data was used for training. Moreover, this dataset has already aligned the images to the same space. If there is motion to the images, does this method still work?

**Detailed Comments:**

It is helpful to add some comments on the limitations if there is any.

**Justification Of Rating:**

The joint undersampling of multimodal images is a reasonable approach to reduce scan time. The join reconstruction using  MIMO CNN algorithm also has better performance than the standard SISO CNN method. This method could potentially be useful in practice with further valiations.

**Paper Type:**

methodological development

**Questions To Address In The Rebuttal:**

None

**Special Issue:**

yes

---

> ### Author Response · Authors · 2020-03-26
> **Discussion/Rebuttal for AnonReviewer2**
>
> We thank the reviewer for the positive feedback.
>
> We find that pixel correspondence between sequences is indeed an important part of this method; as such, with strong motion noise we expect our method to be less effective. On the other hand, there are some minor noise in FLAIR due to long acquisition time; and we still see similar quantitative improvements. The overall effect also depends on the training dataset, and whether certain noise distributions are represented.

---

### Official Review · AnonReviewer1 · 2020-03-13
**This is an interesting paper that jointly learn the k-space sampling and reconstruction from multi-sequence MR image.**

**Rating:** 4
**Confidence:** 5
**Recommendation:** Oral

**Summary:**

This paper works on a task that reconstructs multiple sequence MR images by jointly optimizing sampling and reconstruction. The original formulation for this task is a combinatorial optimization problem since the sampling pattern space is huge. It is transformed to a simpler formulation that showed effectiveness.

**Strengths:**

I overall like the task, formulation and evaluations. The original combinatorial optimization problem is first transformed to be optimization over candidate samples and further formulated as multiple steps method that first learn a general reconstruction network for multi-sequences and select optimal sampling based on this network, followed by fine tuning. The experiments are sufficient and convincing.

**Weaknesses:**

Overall, the idea is interesting and results are good. However, there are still some unclear points as discussed below.

(1) This proposed approach is based on a total time budget, i.e., T_max. Different total time budget may affect the number of samples in each sequence, and affect overall reconstruction accuracies for all sequences. Does the different settings of T_max affect the sampling strategy?

(2) The sampling strategy is learned for each dataset, and tested on the test subset in each dataset. I have concern on generalization ability of this learned  sampling for other datasets of same / different organs.  This is important considering the real application of this approach in MRI imaging.

(3) Please remove some typos, e.g., "not ony " on first page.



**Justification Of Rating:**

The task of multi-sequence MRI is important, the idea that jointly learns sampling and reconstruction is interesting, and the results are overall good. Overall, this is an interesting work deserves to be accepted.

**Paper Type:**

methodological development

**Questions To Address In The Rebuttal:**

I woud like to see some discussions / insights on the questions in weakness.

**Special Issue:**

yes

---

> ### Author Response · Authors · 2020-03-26
> **Discussion/Rebuttal for AnonReviewer1**
>
> We thank the reviewer for the positive feedback on our work. Here is our response for the points of interest that the reviewer listed.
>
> 1. This proposed approach is based on a total time budget, i.e., T_max. Different total time budget may affect the number of samples in each sequence, and affect overall reconstruction accuracies for all sequences. Does the different settings of T_max affect the sampling strategy?
>
> The different settings of T_max do influence the sampling strategy. From our observation, a empirical rule of thumb is that as T_max decreases, the sequence that has the most amount of structural contrast should be sampled more for better recovery. For our real dataset, this tends to be T2. Another point is that FLAIR shows high contrast on things like lesions, tumors, etc., while may not provide the best contrast elsewhere; as such, the sampling strategy should ultimately be decided by radiologists and the intended patient groups.
>
> 2. The sampling strategy is learned for each dataset, and tested on the test subset in each dataset. I have concern on generalization ability of this learned  sampling for other datasets of same / different organs.  This is important considering the real application of this approach in MRI imaging.
>
> We agree with the reviewer that real applications of our method for improvement of all multi-modal MR acquisitions will require datasets that can account for different organs and different acquisition settings, and is indeed essential. More validations are being worked currently, and with proper data engineering, augmentation, and tuning, we find the method to be robust.
>
> 3. Please remove some typos, e.g., "not ony " on first page.
>
> We thank the reviewer for pointing out the typos, and will modify our manuscript accordingly.

---

### Official Review · AnonReviewer4 · 2020-03-14
**Multi-sequence image recovery**

**Rating:** 3
**Confidence:** 5
**Recommendation:** Oral

**Summary:**

This paper develops a method to optimize sampling patterns and reconstruct multiple image sequences in MRI.  The paper makes several approximations to try and obtain a reasonable solution.  The method is evaluated in a comparison between MIMO and SIMO, but there are no comparisons against state of the art methods.

**Strengths:**

Using a BRM to evaluate sampling strategies is a creative idea.

Validating the method using real k-space data is important.

The description is mostly clear.

I don't have other things to say, but need to write something to meet the minimum character count requirements.

**Weaknesses:**

==========================================================================
This paper claims a number of contributions that are not novel and already well known.
==========================================================================

*"We first formulate it as a constrained optimization problem and show that finding the optimal sampling strategy for all sequences and the optimal recovery model for such sampling strategy is combinatorial and hence computationally prohibitive."

This is not a novel contribution because it is already well known that sampling optimization is combinatorial.  Here is a paper from 25 years ago that makes these same observations: S. J. Reeves and L. P. Heck, "Selection of observations in signal reconstruction," IEEE Transactions on Signal Processing, vol. 43, pp. 788--791, March 1995.

*"Our experiments demonstrate that the proposed method outperforms sequence-wise recovery."

This is not a novel observation because there are already many papers that show that multi-sequence reconstruction outperforms sequence-wise reconstruction.  The paper has no comparisons against existing state-of-the-art methods for multi-sequence reconstruction like the methods by Gong or Bilgic.

==========================================================================
There are some methodological concerns.
==========================================================================

*The blind recovery model (BRM) in Section 2.1 is asked to solve a very difficult inverse problem that is much harder than it needs to be.  I imagine that the performance of the BRM is *far* worse than the performance of standard reconstruction methods, which makes me question whether the BRM really provides a good indicator of quality.  The paper only compares MIMO BRM against SISO BRM, but there are no comparisons against standard non-BRM methods.

*"Because BraTS does not provide raw k-space data, we follow common practices (Xiang et al., 2018; Yang et al., 2018) to simulate k-space data."  Unfortunately, it is not possible to generate realistic k-space data like this.  Real MRI images have phase and multiple channels.  Images obtained by taking the Fourier transform of coil-combined magnitude images are much simpler and much easier to reconstruct than real data.  For instance, these images will have zero phase and therefore perfect conjugate symmetry in k-space, which means that half of the samples can be thrown away without compromising reconstruction quality.  This is very different from real MRI data.  This limitation at least needs to be disclosed so that readers are not misled that the simulations are practically meaingful.

"We found that random sampling works better on real data but worse on simulated data."  Seeing a big difference between real data and simulations is a major red flag when the simulations are unrealistic, and suggests that the simulations are not meaningful.

*The paper has access to multi-channel information but does not use it to improve reconstruction.  This is suboptimal and there are no comparisons to methods that would make good use of parallel imaging (like the methods by Gong or Bilgic).

*"We experiment on both low-pass sampling (Xiang et al., 2018) and random sampling (Yang et al., 2018)."

Why not also include uniform undersampling with autocalibration signal, which is the standard approach in parallel imaging?

*It is an oversimplification to assume that imaging time is directly proportional to the number of measured phase encoding lines.  Very often, preparation pulses and additional time are required to get an image into an appropriate steady state. This causes additional time that does not change with the number of phase encoding lines.

==========================================================================
The results are not very good.
==========================================================================

The FLAIR image in Figure 3 is very blurry and is missing clinically-relevant features (e.g., a white matter hyperintensity from the original image is missing in the reconstruction).  This is not useful.

==========================================================================
I also do not think the paper does a good job of describing the literature.
==========================================================================

*"There is a long history of research on how to undersample MR k-space data while maintaining image quality. Lustig et al. (Lustig et al., 2007) first proposed ..."  Undersampling predates Lustig by several decades with origins dating back to at least the 1980s.  An early review article is: Z.-P. Liang, F. E. Boada, R. T. Constable, E. M. Haacke, P. C. Lauterbur, M. R. Smith. "Constrained Reconstruction Methods in MR Imaging," Reviews of Magnetic Resonance in Medicine, vol. 4, pp. 67-185, 1992.

*The citations to deep learning MRI reconstruction methods are highly incomplete and leave out some of the most visible contributions.  There are several recent review articles on deep learning that do a much better job of describing the literature and are worth reading to become more familiar with the state of the field:

C. M. Sandino, J. Y. Cheng, F. Chen, M. Mardani, J. M. Pauly and S. S. Vasanawala, "Compressed Sensing: From Research to Clinical Practice With Deep Neural Networks: Shortening Scan Times for Magnetic Resonance Imaging," in IEEE Signal Processing Magazine, vol. 37, no. 1, pp. 117-127, Jan. 2020.

F. Knoll et al., "Deep-Learning Methods for Parallel Magnetic Resonance Imaging Reconstruction: A Survey of the Current Approaches, Trends, and Issues," in IEEE Signal Processing Magazine, vol. 37, no. 1, pp. 128-140, Jan. 2020.

D. Liang, J. Cheng, Z. Ke and L. Ying, "Deep Magnetic Resonance Image Reconstruction: Inverse Problems Meet Neural Networks," in IEEE Signal Processing Magazine, vol. 37, no. 1, pp. 141-151, Jan. 2020.

*"As shown in (Xiang et al., 2018; Huang et al., 2012), there exists a strong correlation between sequences of the same patient, as they share the underlying anatomical structures."

This correlation has been used in much earlier literature.  See

Leahy R, Yan X. Incorporation of anatomical MR data for improved functional imaging with PET. Information processing in medical imaging 1991. pp 105–120.

Webb, A.G., Liang, Z.-P., Magin, R.L. and Lauterbur, P.C. (1993), Applications of reduced-encoding MR imaging with generalized-series reconstruction (RIGR). J. Magn. Reson. Imaging, 3: 925-928. doi:10.1002/jmri.1880030622

Haldar, J.P., Hernando, D., Song, S.-K. and Liang, Z.-P. (2008), Anatomically constrained reconstruction from noisy data. Magn. Reson. Med., 59: 810-818. doi:10.1002/mrm.21536


==========================================================================
There are also some notation issues that are not clear.
==========================================================================

*The paper should define the meaning of the variable theta.

**Justification Of Rating:**

This paper has a number of limitations, but I think these can all be addressed if the authors are responsive to comments.  The method itself is creative and can be thought-provoking even if it is not state-of-the-art.  But it's necessary to list all of the limitations of the work to avoid making readers think the method is more mature than it really is.

**Paper Type:**

methodological development

**Questions To Address In The Rebuttal:**

The most important issues to address are:

1. Make sure that the contribution is clear and that you are not taking credit for well-known existing ideas.

2.  Include some kind of comparison against a non-BRM method.

3. Either remove the unrealistic simulation results or make sure that readers are aware that these results are very unrealistic and may not be relevant to real applications.

4. Acknowledge the poor performance of the method for FLAIR reconstruction

**Special Issue:**

no

---

> ### Author Response · Authors · 2020-03-26
> **Discussion/Rebuttal for AnonReviewer4**
>
> We would like to first thank the reviewer for the thoughtful and detailed comments. Many of these comments are very helpful for improving the quality of our work, here is our response to some of the questions.
>
> 1. Make sure that the contribution is clear and that you are not taking credit for well-known existing ideas.
>
> We recognize the fact of sampling optimization being combinatorial in general is both intuitive and well-known. However, we find it still helpful for motivation to formulate such problem in the context of multi-sequence k-space acquisition, which we have not found to be made elsewhere. We agree with the reviewer that an improved writing with more citation will lead to less misunderstanding for readers, and will modify the manuscript accordingly.
>
> 2. Include some kind of comparison against a non-BRM method.
>
> The reviewer commented that "The blind recovery model (BRM) in Section 2.1 is asked to solve a very difficult inverse problem that is much harder than it needs to be. I imagine that the performance of the BRM is *far* worse than the performance of standard reconstruction methods, which makes me question whether the BRM really provides a good indicator of quality.  The paper only compares MIMO BRM against SISO BRM, but there are no comparisons against standard non-BRM methods."
>
> There seems to be some misunderstanding here. For fair comparisons, we compared the performance of BRM to reconstruction models of the same structure that are trained on specific sampling masks (named "Dedicated" in the legend of Figure 2). We showed in Figure 2 (on the right) that while there is a performance gap between the non-BRM setting and the BRM setting, BRM is highly correlated with the recovery quality of the non-BRM setting.
>
> 3. Either remove the unrealistic simulation results or make sure that readers are aware that these results are very unrealistic and may not be relevant to real applications.
>
> We agree with the reviewer that explaining the difference between simulated and real data in more details will help readers interpret the results, and will modify accordingly.
>
> 4. Acknowledge the poor performance of the method for FLAIR reconstruction
>
> We thank the reviewer for pointing out that the visual quality of FLAIR seems inferior, and seek to clarify here. In the dataset that we collected, FLAIR is the most noisy of the three sequences, most likely due to its much longer acquisition time (the noise is more observable by magnifying the HR images). As a result, the visual quality of FLAIR reconstruction is also lower compare to other sequences, as the CNN attempts to average out the noise and simultaneously washes out some of the details under L1 loss function. Looking from the perspective of quantitative comparison, however, the reconstructed FLAIR image through our method improves against baseline similarly, which is an indication to us that the problem is more about the data quality than the methodology. We agree that a better clarification should be made here to explain the lower visual quality of FLAIR images to help reader better understand.

---

### Meta-Review · Area_Chair1 · 2020-03-31
**MetaReview of Paper18 by AreaChair1**

**Rating:** 3
**Recommendation For Accepted Papers:** Poster

**Metareview:**

The paper proposes to recontruct multi-sequence MR image from undersampled k-space data. Using a blind recovery model to evaluate sampling strategies is a creative idea. But some related work has to be cited. For example, the idea of employing deep learning for multicontrast MRI imaging was explored in  "Feasibility of Multi-Contrast MR Imaging Via Deep Learning in ISMRM 2017"

**Paper Type:**

methodological development

**Special Issue:**

no

---

### Decision · Program_Chairs · 2020-04-11

Accept